# Freewater estimatoR using iNtErpolated iniTialization (FERNET): Characterizing peritumoral edema using clinically feasible diffusion MRI data

**Drew Parker[1,2,3], Abdol Aziz Ould Ismail[1,2,3], Ronald Wolf[3], Steven Brem[4], Simon Alexander[5], Wes Hodges[5], Ofer Pasternak[6], Emmanuel Caruyer[7], Ragini Verma[1,2,3,4]***

**1** DiCIPHR (Diffusion and Connectomics in Precision Healthcare Research) Lab, Perelman School of Medicine, University of Pennsylvania, Philadelphia, PA, United States of America, **2** Center for Biomedical Image Computing and Analytics (CBICA), Perelman School of Medicine, University of Pennsylvania, Philadelphia, PA, United States of America, **3** Department of Radiology, Perelman School of Medicine, University of Pennsylvania, Philadelphia, PA, United States of America, **4** Department of Neurosurgery, Perelman School of Medicine, University of Pennsylvania, Philadelphia, PA, United States of America, **5** Synaptive Medical Inc., Toronto, ON, Canada, **6** Departments of Psychiatry & Radiology, Brigham and Women's Hospital, Harvard Medical School, Boston, MA, United States of America, **7** IRISA UMR, Univ Rennes, CNRS, Inria, Inserm, Brittany, France

* ragini@pennmedicine.upenn.edu, ragini.verma@uphs.upenn.edu

**Data Availability Statement:** We have uploaded the data set necessary to replicate our study findings. The healthy controls used in the paper are

## Abstract

Characterization of healthy versus pathological tissue in the peritumoral area is confounded by the presence of edema, making free water estimation the key concern in modeling tissue microstructure. Most methods that model tissue microstructure are either based on advanced acquisition schemes not readily available in the clinic or are not designed to address the challenge of edema. This underscores the need for a robust free water elimination (FWE) method that estimates free water in pathological tissue but can be used with clinically prevalent single-shell diffusion tensor imaging data. FWE in single-shell data requires the fitting of a bi-compartment model, which is an ill-posed problem. Its solution requires optimization, which relies on an initialization step. We propose a novel initialization approach for FWE, FERNET, which improves the estimation of free water in edematous and infiltrated peritumoral regions, using single-shell diffusion MRI data. The method has been extensively investigated on simulated data and healthy dataset. Additionally, it has been applied to clinically acquired data from brain tumor patients to characterize the peritumoral region and improve tractography in it.

## 1. Introduction

Diffusion tensor imaging (DTI), one of the basic frameworks of diffusion MRI (dMRI), is frequently used in clinical studies to characterize tissue microstructure and provides orientation information for delineation of basic fiber tracts using tractography. DTI can be used to enrich

part of the Philadelphia Neurodevelopmental Cohort dataset which is available at https://www.ncbi.nlm.nih.gov/projects/gap/cgi-bin/study.cgi?study_id=phs000607.v3.p2. The healthy controls from Dataset 1 and simulated data can be found at https://github.com/DiCIPHRLab/ Fernet/tree/paper_data. The brain tumor data in the paper is restricted. It requires an agreement with Penn, and can only be released once the said had been executed. The contact person for that isRagini Verma, ragini@pennmedicine.upenn.edu.

**Funding:** This research was supported by National Institutes of Health (NIH) grants R01NS096606 (PI: Ragini Verma) and R01MH108574 (PI: Ofer Pasternak), and research grant from Synaptive Medical 30071788 (PI: Ragini Verma). Synaptive Medical provided support in the form of percent effort towards salaries for authors [RV, DP, AAOI]. They were involved in design of experiments, planning analysis, and application to brain tumor data. They contributed to writing of the manuscript.

**Competing interests:** Synaptive Medical provided support in the form of percent effort towards salaries for authors [RV, DP, AAOI]. This affiliation with Synaptive Medical does not alter our adherence to PLOS ONE policies on sharing data and materials.

radiomic markers of pathology and inform studies of connectivity in the brain, and is essential for neurosurgical and treatment planning of some brain tumors [1]. Since dMRI relies on the measurement of water displacement in tissue, it is confounded by disease processes that cause accumulation of water in the extra-cellular space, such as vasogenic edema, infiltrative edema or other parenchymal pathology (e.g. traumatic brain injury) which can alter the diffusion of water. These processes compromise the specificity of diffusion indices characterizing white matter, such as fractional anisotropy (FA) and mean diffusivity (MD), and subsequently tractography, leading to interpretation errors in clinical studies, including those tailored for surgical and treatment planning of brain tumors.

These issues can be alleviated by an accurate non-invasive estimation and compartmentalization of water (and its restriction by presence of pathology) in the brain by multi-compartment modeling of the diffusion data. Such modeling captures the healthy tissue as a compartment, and the contaminants (effects of disease) as one or more compartments. Multi-compartment modeling is able to alleviate partial volume effects and allows for more accurate diffusion indices, improved tractography, and provides novel contrasts such as volume fraction that capture underlying pathophysiological processes [2]. Although the importance of multi-compartment modeling has been recognized [3–7], most of these use multi-shell diffusion acquisitions, since the additional shells allow for more robust and accurate estimation of model parameters. Such acquisitions, however, remain clinically challenging at this time, owing to the length of scan being limited on a patient and inherent limitations of clinical scanners and surgical planning software

Single-shell DTI is the predominant type of diffusion MRI acquired in a busy clinical setting. Additionally, large cancer databases, such as the Adult Brain Tumor Consortium (ABTC) (http://www.abtconsortium.org) have single-shell DTI data only. The ability to interrogate the peritumoral manifestation of free water in such large datasets, with their wealth of clinical data, including pharmacological and surgical treatment responses, quality of life measures, and survival rates, could be crucial in assessing neoplastic infiltration and edema and potentially provide new radiomic features for oncological evaluation. This highlights the need for free water estimation that is accurate in the presence of peritumoral edema, using clinically realistic acquisitions.

The fitting of a bi-compartment model to single-shell data is a mathematically ill-posed problem [8] with infinitely many solutions. The earliest attempts of fitting a bi-compartment model relied on multiple shells [9, 10]. Pasternak *et al.* [2] extended the model fit to single-shell acquisitions by positing that the problem could be addressed by an appropriate initialization of the model parameters and by a spatial regularizer that stabilizes the fit. This free water estimation allowed for a better reconstruction of healthy fornix tracts and enabled fiber tracking through edema. The application of this model has led to significant findings in various applications, including depression [11], Parkinson's disease [12–14] and schizophrenia [15]. The initialization, however, is affected by scanner inhomogeneity [16] and it underestimates the free water in healthy tissue while producing physiologically implausible diffusion indices in the peritumoral region [17]. It has also not been extensively validated in simulated and brain tumor data.

The overarching goal of this paper is to present a paradigm for free water elimination in peritumoral tissue using clinically acquired single-shell diffusion data. This is achieved by fitting a bi-compartment model, extending the work in [2] with a novel interpolated initialization designed to work optimally in edematous and healthy regions. The two compartments describe the underlying tissue and edema. This free water estimation paradigm will subsequently be referred to as FERNET (Freewater EstimatoR using iNtErpolated iniTialization). In this paper, FERNET is tested on simulated DTI data with varying free water fractions,

anisotropy levels, and underlying diffusivities, and clinically acquired data of healthy controls and tumor patients with edema.

## 2. Methods

### 2.1. Overview

We describe a novel initialization approach, aiming to improve free-water estimation from clinically acquired single-shell data. We present comprehensive experiments on both simulated and real data, in both healthy tissue as well as in peritumoral regions, demonstrating the performance of our method compared to existing ones in both healthy and peritumoral tissue.

### 2.2. Freewater EstimatoR using iNtErpolated iniTialization (FERNET)

Following the bi-tensor approach suggested previously [2, 10], we model the diffusion signal in each voxel with two compartments: a tensor compartment representing the underlying tissue and an isotropic compartment with a fixed diffusivity equal to the diffusivity of free water in biological tissues. This is mathematically modeled as

$$A_i(D, f) = f\, e^{-bq_i^T D q_i} + (1 - f)\, e^{-bd} \tag{1}$$

where the first and second terms model the tissue and free water compartment, respectively, $A_i$ is the signal attenuation of the diffusion weighted image acquired along the $i^{th}$ gradient direction, $f$ is the tissue volume fraction, $b$ is the magnitude of diffusion weighting, $q_i$ is $i^{th}$ gradient direction, $D$ represents the diffusion tensor used for modeling the tissue compartment, and $d$ is the diffusivity in the isotropic compartment, which is fixed at 3.0 x $10^{-3}$ mm$^2$/s. Fitting this model using a single-shell dMRI acquisition is a problem with infinitely many solutions [2] Finding a solution to such a problem requires a combination of a good initialization and optimization. We use a novel initialization designed to address both healthy and pathological tissue followed by gradient descent to fit the model. Specifically, we propose the following initialization of the tissue volume fraction:

$$f_{init} = f_{b=0}^{1-\alpha} \cdot f_{MD}^{\alpha} \tag{2}$$

$f_{init}$ is a logarithmic interpolation between two initialization strategies, one denoted as $f_{b=0}$ and the other as $f_{MD}$.

The first strategy, $f_{b=0}$, is similar to that proposed in [2], based on scaling the mean unweighted image ($S_0$) with respect to representative unweighted signals of WM ($S_t$) and CSF voxels ($S_w$):

$$f_{b=0} = 1 - \frac{\log\left(S_0/S_t\right)}{\log\left(S_w/S_t\right)} \tag{3}$$

The resulting map $f_{b=0}$ is further constrained by estimates of the minimum and maximum tissue fraction corresponding to minimum and maximum expected eigenvalues (i.e., diffusivities) of a tensor modeling the brain tissue, given by

$$f_{min} = \frac{\min(\hat{A}) - e^{-bd}}{e^{-b\lambda_{max}} - e^{-bd}}, \; f_{max} = \frac{\max(\hat{A}) - e^{-bd}}{e^{-b\lambda_{min}} - e^{-bd}} \tag{4}$$

where parameters $\lambda_{min}$ and $\lambda_{max}$ are set to 0.1x10$^{-3}$ mm$^2$/s and 2.5x10$^{-3}$ mm$^2$/s, respectively, and $\hat{A}$ is the vector of signal attenuation, *i.e.* the values of the diffusion weighted images

divided by $S_0$. It may be noted that these parameters represent a putative range of radial and axial diffusivity, respectively, in WM and were taken from [2].

There are two differences in our proposed estimation of $f_{b=0}$ from the previously reported method for free water elimination. First, we define $S_t$ as the 5th percentile of the unweighted signal in a region of WM, and we define $S_w$ as the 95th percentile of unweighted signal in a region of CSF. Previously, these values were defined as the mean unweighted signal within the two regions. Second, initial values outside the range $[f_{min}, f_{max}]$ are set to the nearest of the two values. Previously, the estimated value was replaced with the value $\frac{1}{2}(f_{min} + f_{max})$ in this situation.

The second strategy, $f_{MD}$, in equation 2 is given by

$$f_{MD} = \frac{e^{-bMD} - e^{-bd}}{e^{-bMD_{tissue}} - e^{-bd}} \tag{5}$$

where MD is the mean diffusivity from the standard tensor fit in a voxel of interest, and $MD_{tissue}$ is a fixed value ($0.60 \times 10^{-3}$ mm$^2$/s [18, 19]) representing the expected MD of a WM voxel that is not impacted by partial voluming or pathology.

Finally, the value of $\alpha$ in Eq 2 is set to the value of $f_{b=0}$, but constrained to the range $[0,1]$ rather than the range $[f_{min}, f_{max}]$. This results in an interpolation of the two strategies which depends on $S_0$ in the voxel of interest. For example, in a WM voxel with normal appearing T2 signal, where the $S_0$ is close to the value $S_t$, the value of $\alpha$ is nearly 1 and the result of this interpolation is closer to that using $f_{MD}$. In regions that appear like CSF in the T2 contrast, the value of $\alpha$ is nearly 0 and the result is closer to $f_{b=0}$. In voxels with intermediate T2 intensity, which include vasogenic edema, the peritumoral region, and voxels with partial volume of CSF, this interpolation is closer to the geometric mean of the two approaches. In this way, our initialization is designed to smoothly modulate its behavior based on the signal properties of the voxels without relying on any segmentation beyond the selection of representative CSF and WM regions.

As the proposed method depends on the $S_0$ in each voxel, any bias in that image resulting from inhomogeneous magnetic field in the scanner will likely lead to a biased $f_{init}$ (as well as a biased $f_{b=0}$). Therefore, we propose the use of bias-field-corrected maps of $S_0$ for free water elimination problems.

In summary, our interpolated initialization method (FERNET) differs from the approach in [2], which we subsequently call "b0 initialization", in three key aspects:

- A novel initialization strategy that aims to improve the estimation of the free water compartment in both healthy and pathological tissue.

- Bias field correction of $S_0$ before initialization.

- The initial fraction values outside of the plausible range are replaced with the nearest plausible value rather than the mean before optimization.

## 2.3. Human datasets

This study was reviewed and approved by the institutional review board of the University of Pennsylvania. The following datasets will be used in the different experiments. Written informed consent was obtained from each of the participants in the studies.

**Dataset 1.** 30 participants were selected from an ongoing tumor study, including 21 healthy controls, and 9 patients with a diagnosis of glioblastoma. All participants underwent a multi-shell diffusion acquisition with b-values of 300 (15 directions), 800 (30 directions), and

2000 s/mm$^2$ (64 directions) and 9 unweighted volumes. The data were acquired using a Siemens 3T TIM Trio scanner with a 32-channel head coil and echo planar (EP) sequence with TR = 5216 ms, TE = 100 ms, at a spatial resolution of 2 x 2 x 2 mm. T1, T2, FLAIR, and T1 contrast-enhanced (T1CE) scans were also acquired for the 9 brain tumor patients.

**Dataset 2.** This dataset comprised 216 healthy controls from the Philadelphia Neurodevelopmental Cohort (PNC) [20]. The age range of these selected subjects was 18 to 23 years. There were 118 females and 98 males. The data were acquired on a 3T Siemens TIM Trio scanner, using a 32-channel head coil and a twice-refocused spin-echo (TRSE) single-shot EPI sequence with TR = 8100 ms and TE = 82 ms, at a b-value of 1000 s/mm$^2$ for a total 64 weighted diffusion images and seven unweighted images. The acquired spatial resolution was 1.875 x 1.875 x 2 mm.

**Dataset 3.** This dataset comprised 143 brain tumor patients (89 glioblastoma/54 metastasis), putatively representing a range of levels of tumor cell infiltration and extent of edema. The age range of this data was 19 to 87 years and there were 77 females and 66 males. Single-shell diffusion data were acquired on Siemens 3T Verio scanner with TR = 5000ms and TE = 5000/86ms, 3 unweighted volumes, and 30 diffusion weighted volumes at a b-value of 1000 s/mm$^2$. The acquired spatial resolution was 1.72 x 1.72 x 3 mm. Additionally, T1, T2, FLAIR, and T1CE were acquired.

**Data preprocessing.** All datasets were investigated visually for quality assurance (QA). The datasets were denoised [21] and corrected for eddy current-induced distortions [22]. In *Dataset 3*, voxels were resampled to 2mm isotropic resolution. The b0 images were extracted and skull-stripped using FSL's BET tool [23]. Bias correction of the b0 images was performed with N4BiasFieldCorrection from the Advanced Normalization Tools (ANTs) package. A DTI model was fit within the brain mask using a weighted linear least-squares method. FA and MD maps of healthy controls in *Dataset 1* and *Dataset 2* were non-linearly registered to the JHU-MNI-ss ("Eve") atlas [24] using the SyN registration algorithm from the ANTs package. Automatic tumor and peritumoral region segmentation was obtained in the patients from *Dataset 1* and *Dataset 3* using GLISTR [25], after co-registering the T1, T2, FLAIR and T1CE images using FSL's flirt tool.

## 2.4. Experiments on simulated data

To address the lack of ground truth, especially in edematous and infiltrated regions, we simulated data with varying ground truth mean diffusivities, anisotropies, and free water volume fractions. These simulated datasets follow a bi-compartment model where one of the compartments represents tissue and the other is isotropic with a fixed diffusivity (3.0 x 10$^{-3}$ mm$^2$/s). The simulation of the weighted images was obtained using the bi-tensor model available in Dipy's multi-tensor simulator [26]. For simulating an unweighted (b0) image, $S_0$, we calculated the transverse magnetization of a spin-echo experiment as a linear sum of the contribution of each compartment as proposed in *Phantomas* [27]:

$$S_0 = f_{WM}\rho_{WM}\left(1 - e^{\frac{-TR}{T1_{WM}}}\right)e^{\frac{-TE}{T2_{WM}}} + f_{CSF}\rho_{CSF}\left(1 - e^{\frac{-TR}{T1_{CSF}}}\right)e^{\frac{-TE}{T2_{CSF}}} \qquad (6)$$

where $f_{WM}$ is the volume fraction of white matter, $f_{CSF}$ is the volume fraction of CSF, $\rho$ is the proton density, and TR/TE are repetition and echo times. The first term of Eq 6 represents the signal from WM, while the second term represents the signal from CSF. Although values of $\rho$, T1 and T2 of the two tissue types are not known *a priori*, they do not change when volume fraction is varied. Thus, we simulated $b_0$ for each voxel as:

$$S_0 = fS_{0_{WM}} + (1 - f)S_{0_{CSF}} \qquad (7)$$

where $f$ is the desired volume fraction of tissue, and $S_{0_{WM}}$ and $S_{0_{CSF}}$ are reference values taken from human data. The diffusion signal, $S_i$, for the $i^{th}$ gradient direction is simulated as:

$$S_i = S_0 \left( f\, A_{tissue} + (1-f)A_{water} \right) \tag{8}$$

where $A_{tissue}$ and $A_{water}$ represent the attenuation of tissue and water, respectively. Thus, this model can simulate white matter tissue with ground truth diffusion tensor eigenvalues and any degree of additional free water.

Using this synthetic data model, we devised three simulations based on human diffusion MRI data: two from the WM of healthy controls to represent healthy WM tissue, and one derived from a tumor dataset to represent tumor cell infiltration. Various values of volume fraction were then added to these simulations to represent a range of edema. To obtain representative diffusion MRI measurements of healthy WM, 21 healthy controls of *Dataset 1* were registered to the JHU-MNI-SS template, where the maps of their three tensor eigenvalues and B0 images were averaged. Masks of WM and CSF were created in the template space via segmentation of the template T1 image, and eroded by 1 voxel each to eliminate partial voluming between tissue types. Mean values of B0 signal intensity and the three eigenvalues were calculated within these masks, resulting in an FA of 0.5 and MD of 7.7x10$^{-4}$ mm$^2$/s. These values, along with a ground truth VF of 0, formed the basis of the first simulation of healthy WM (Fig 1A), to which simulated volume fraction was added to represent edema. In a second scenario, healthy WM could be considered to have some unknown level of extracellular water, and thus the measured eigenvalues likely do not correspond to a VF of exactly 0. In order to model this scenario, a simulation of healthy WM was created where we assigned the measured eigenvalues to an arbitrary non-zero VF of 0.15 and extrapolated to a VF of 0 following Eq 8. The resulting FA was 0.6 and MD was 6.0x10$^{-4}$ mm$^2$/s, forming the basis of the second simulation of healthy WM (Fig 1B). Third, to form a synthetic representation of tumor cell infiltration and extracellular edema, a tumor patient from *Dataset 1* was selected and a mask of restricted diffusion (due to increased cellularity in the tumor) was hand-drawn on the MD image (Fig 1C). The measured average FA was 0.1 and MD was 5.5x10$^{-4}$ mm$^2$/s. In each simulated scenario, we added a free water compartment with ten levels of volume fraction from 0 to 0.9 to represent edema, and ten levels of Rician noise (signal to noise ratio (SNR) of 10 to 100), realizing 100 simulations per noise level. In addition, for every simulated tensor, we rotated the tensor directions randomly 100 times. Thus, there were 1,000,000 simulated tensors and free water compartments for each of the three simulations. Free water estimation error was computed in each simulation as the difference between ground-truth free water and estimated free water. The estimation errors of FERNET and b0 initialization were compared.

## 2.5. Experiments on human data

**2.5.1. Comparison of single-shell and multi-shell free water estimations.** Since ground truth values of the free water volume fraction in human data are not available, we estimated free water volume fraction through a multi-shell method [28] applied to all the shells available in *Dataset 1*, to use as a reference standard. This is based on the assumption that multi-shell acquisitions provide a better estimation of free water. This method was chosen because it produces two compartments just like our method but uses multi-shell data to alleviate the issues with single-shell. We extracted the b = 800 s/mm$^2$ shell to obtain single-shell data on which the two initialization approaches for free water estimation were applied. Two analyses were performed. First, the single-shell methods were compared to the reference standard by calculating voxel-wise correlation coefficient and mean square error, using the 21 healthy controls co-registered to the same atlas. The second analysis was performed in the peritumoral regions of the

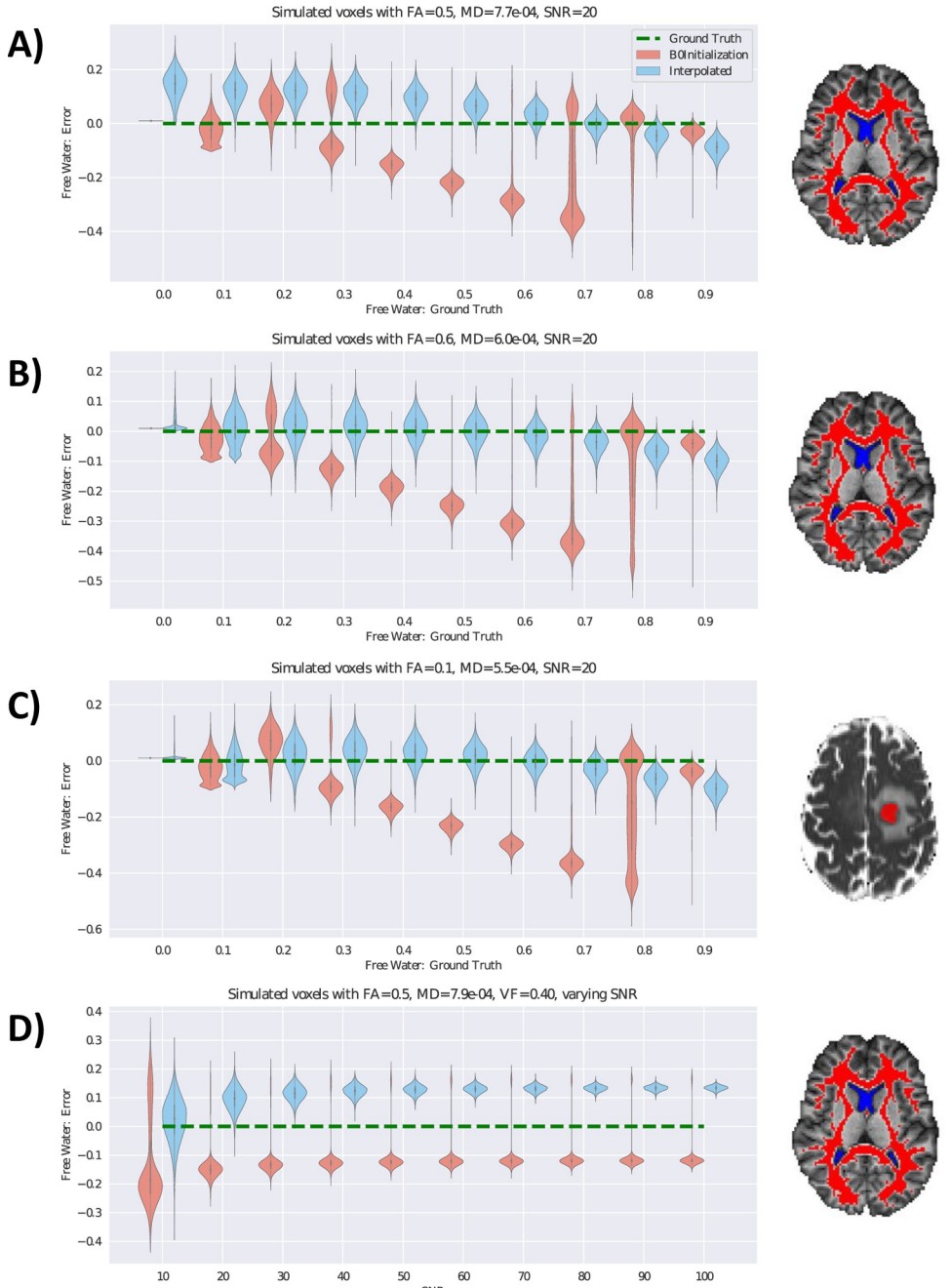

**Fig 1. Free water estimation on simulated data.** The error in free water estimation using simulated data with varying ground truth volume fraction is presented using violin plots. A) Eigenvalues averaged across WM voxels of 21 healthy controls were used to generate synthetic diffusion data with volume fraction of 0. Average MD was $7.7 \times 10^{-4}$ and average FA was 0.5. Synthetic free water representing vasogenic edema was added to this baseline. The masks of WM (red) and CSF (blue) used to calculate these values in 21 healthy controls are shown to the right. B) Healthy WM in human data has some unknown, but non-zero, free water fraction. Average values from (A) were assigned to a VF value of 0.15 and extrapolated to find new values of MD and FA which correspond to a VF of 0. MD in this scenario was $6.0 \times 10^{-4}$ and FA was 0.6. C) Eigenvalues were averaged across the voxels of one tumor patient within the region of restricted diffusion in the tumor (shown in red to the right). The average FA was 0.1 and MD was $5.5 \times 10^{-4}$. These were used to generate synthetic diffusion data with VF of 0, and synthetic free water was added to this baseline. Experiments A-C have an SNR of 20, with free water varying from 0 to 0.9. The y-axis represents the difference between the free water estimation and the corresponding ground truth of free water (x-axis). Green dotted lines in each plot represent the zero error in free water estimation. Violin plots of the two methods are staggered for clarity. Results show that FERNET initialization is more accurate in estimating free water than the b0 initialization where FW>0.3, and

FERNET initialization does not have discontinuities in predicted free water volume fraction. D)shows the effect of SNR on the error in free water, where SNR varies from 10 to 100. The plotted results are from the simulated data used in (A) with VF = 0.4. SNR has an effect on the variance of error in free water, as well as a small effect on the mean error in free water.

9 tumor patients to compare the estimated free water of the two single-shell approaches against the reference standard.

**2.5.2. The effect of regularization on free water estimation..**   In order to demonstrate that our proposed initialization alleviates the dependence on regularization, we compared the histogram of the free water values in the whole brain as well as in the peritumoral region with and without the use of regularization for the two initialization approaches in a tumor patient from *Dataset 3*. The regularization was implemented using an in-house MATLAB tool.

**2.5.3. The effect of bias field correction and constraints on free water estimation.**   As shown in Eq 3, the initialization depends on representative b0 signals in both tissue and free water compartments as well as the b0 signal itself. We applied a bias field correction [29] on the b0 image, and compared results of free water estimation with and without the application of bias correction. Analysis was performed using co-registered data from *Dataset 2*. At every voxel, the percentage of controls with artifactual initial estimates of the free water corrected tensor, defined as MD $< 0.40x10^{-3}$ mm$^2$/s [30], was calculated.

**2.5.4. Free water estimation in brain tumor data.**   We selected two tumor patients from *Dataset 3*: Patient 1 with a metastatic tumor and Patient 2 with glioblastoma. These represent case studies for investigating the results of the two approaches. The b0 initialization method was performed with spatial regularization, while FERNET was not. We visually compared the free water and corrected FA of the peritumoral region to healthy tissue in the contralateral hemisphere.

**2.5.5. Tractography.**   Tractography was performed on *Dataset 3* (143 brain tumor patients) using Diffusion Toolkit [31], on tensor images calculated with and without FERNET free water elimination. Tensors calculated without free water elimination are referred to as the "standard" tensor fit. The second order Runge-Kutta algorithm was used for tracking, with an angle threshold of 45$^o$, a step size of 1 mm, and an FA threshold of 0.2. Five bundles of interest (corticospinal tract, inferior longitudinal, inferior fronto-occipital, uncinate and arcuate fasciculi) in each hemisphere were extracted from each tractogram, using the RecoBundles algorithm [32], with a pruning parameter of 7 mm. Finally, the "edema coverage", defined as the percentage of voxels in the peritumoral edema region containing one or more streamlines belonging to any of the ten bundles of interest, was computed for every patient with and without free water elimination. A percentage difference was calculated relative to the edema coverage of the tractography without free water elimination.

## 3. Results

### 3.1. Free water estimation on simulated data

Fig 1 presents the results of comparing free water estimation on three simulations detailed in Section 2.4: healthy WM, with added levels of free water (representing edema) (Fig 1A), an extrapolated scenario of healthy WM containing some free water (Fig 1B), and a region of restricted diffusion with varying degrees of edema from one tumor patient (Fig 1C). Our findings show that in the presence of simulated edema (FW>0.3), FERNET improved the estimation of the free water across the three scenarios (Fig 1). The FERNET estimation had a similar variance to the b0 initialization at low values of ground truth VF, but did not display the heavily skewed distribution of errors of the b0 initialization method, as shown by the spread in

the violin plots, especially at higher values of ground truth VF. The FERNET estimation did not show signs of a bimodal distribution in errors. These experiments were performed at a simulated SNR of 20. Fig 1D shows the distribution of errors for the simulated healthy WM at a ground truth VF of 0.4 as SNR was varied from 10 to 100. The mean bias of both estimates changed slightly from SNR = 10 to SNR = 30, after which the variance in error diminished with increasing SNR. The b0 initialization method demonstrated a skewed distribution of errors at every noise level.

### 3.2. Free water estimation on human data

Fig 2 shows the results of comparing the free water estimation using FERNET and the b0 initialization with the free water estimation obtained from the multi-shell method used as a reference standard in the absence of ground truth, as detailed in Section 2.5. Voxel-wise maps of correlation coefficient and mean square error (MSE) of the free water fraction between single-shell methods and the multi-shell method (Fig 2A for the b0 initialization, Fig 2B for the interpolated initialization of FERNET) showed a weaker correlation between the free water maps, especially in WM regions, and a relatively higher MSE in the b0 initialization when compared to FERNET. These correlation findings were consistent with Fig 2C, where the free water values in WM of 21 healthy controls using FERNET were more aligned with the identity line when compared to the multi-shell reference standard. The correlation between b0 initialization and multi-shell estimation was 0.59, and the correlation between FERNET and multi-shell estimation was 0.81. The correlation of free water values in the peritumoral regions of 9 patients (Fig 2D) between the b0 initialization and the reference standard was 0.60, and between FERNET and the reference standard was 0.75.

### 3.3. The effect of regularization on free water estimation

Fig 3 shows the behavior of the free water estimation approaches to regularization in two brain tumor patients. The effect of regularization is assessed via the difference maps of the free water volume fraction obtained with and without regularization, in the peritumoral area and the whole brain. The histograms of the free water estimation in the whole brain and in the peritumoral region show a closely matched free water estimation for regularized and non-regularized approaches when using FERNET.

### 3.4. Effect of bias field correction

Fig 4 shows the results of investigating the use of bias field correction in free water elimination. It shows the percentage of participants that yielded an artifactual fit ($MD < 0.40 \times 10^{-3}$ mm$^2$/s) in the initial corrected tensor (before the optimization phase) at each voxel in the brain, without (Fig 4A) and with (Fig 4B) bias field correction. Without bias field correction, the corrected tissue tensor is initialized with indices that are physiologically implausible. Such diffusivity values are estimated in a large percentage of controls in the peripheral regions of the cortex and cerebellum, areas which are most impacted by bias.

### 3.5. Free water elimination in brain tumor data

Fig 5 shows the free water volume fraction and the corrected FA maps using the two free water estimation methods on two brain tumor patients: Patient 1 with metastatic tumor and Patient 2 with glioblastoma. FERNET free water maps had spatially smoother contrast than those from the b0 initialization, and FA values in the WM ipsilateral to the tumor matched the FA of the contralateral WM better, especially in Patient 1.

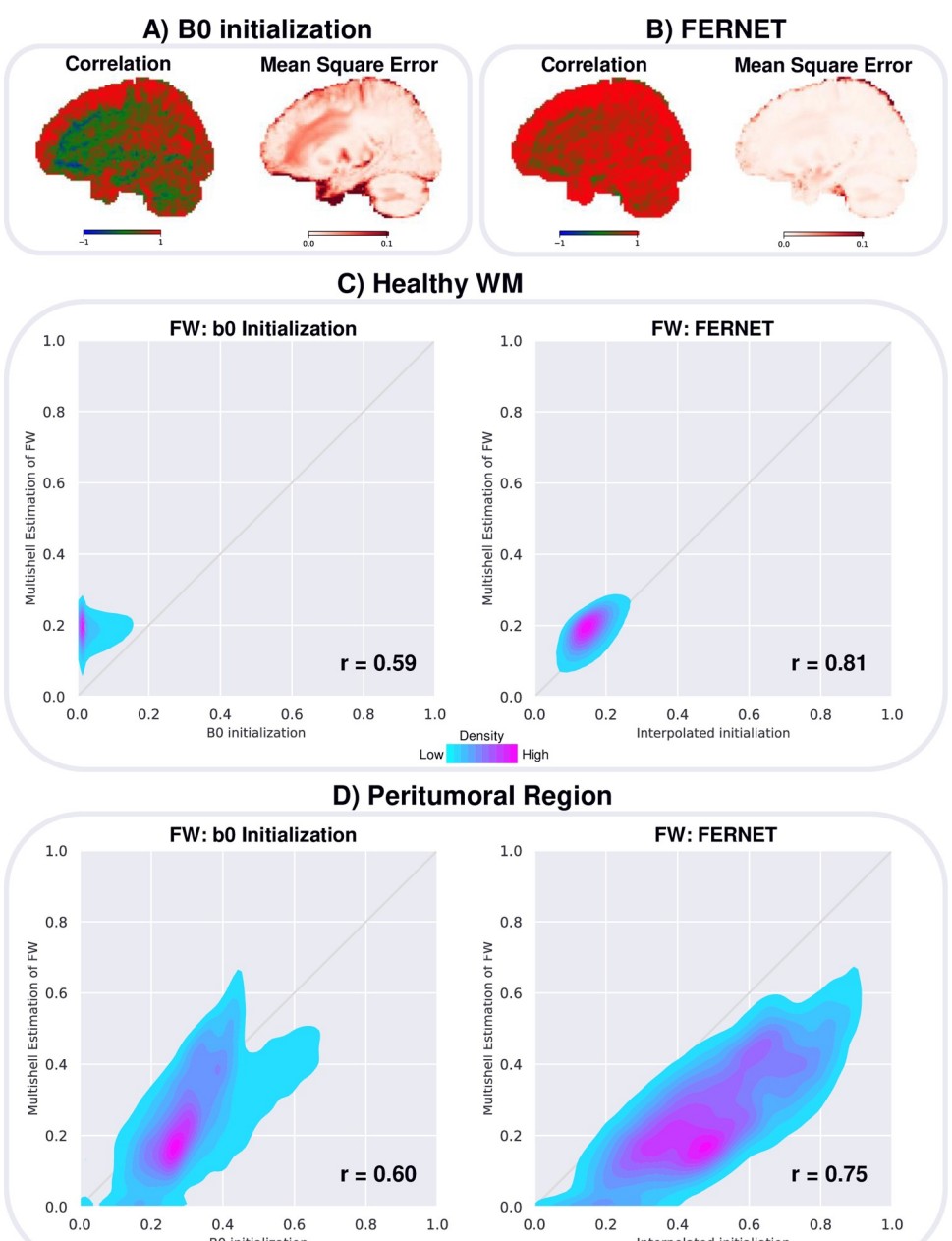

**Fig 2.** Free water estimation in human data compared to the multi-shell reference standard correlation coefficient and mean square error (MSE) are measured at every voxel for 21 healthy controls to assess the contrast and difference between the free water (FW) map derived from A) b0 initialization and B) interpolated initialization (FERNET) compared to the FW map derived from multi-shell. Correlations are displayed with a blue-green-red colormap where blue is -1, green is 0 and red is 1. More red voxels indicate a stronger agreement with the reference standard. MSE are displayed with a colormap from white to dark red. Lighter voxels indicate a stronger agreement with the reference standard. C) shows the scatterplots of free water fraction values within the WM of 21 controls, demonstrating that the proposed interpolated initialization is more aligned with the reference standard in WM. D) shows the scatterplots of free water fraction in the peritumoral regions of 9 brain tumor patients, demonstrating that neither of the single-shell methods obtained an estimation in the peritumoral region that is perfectly consistent with reference standard. However, correlations are higher with FERNET (r = 0.75) than with b0 initialization (r = 0.60).

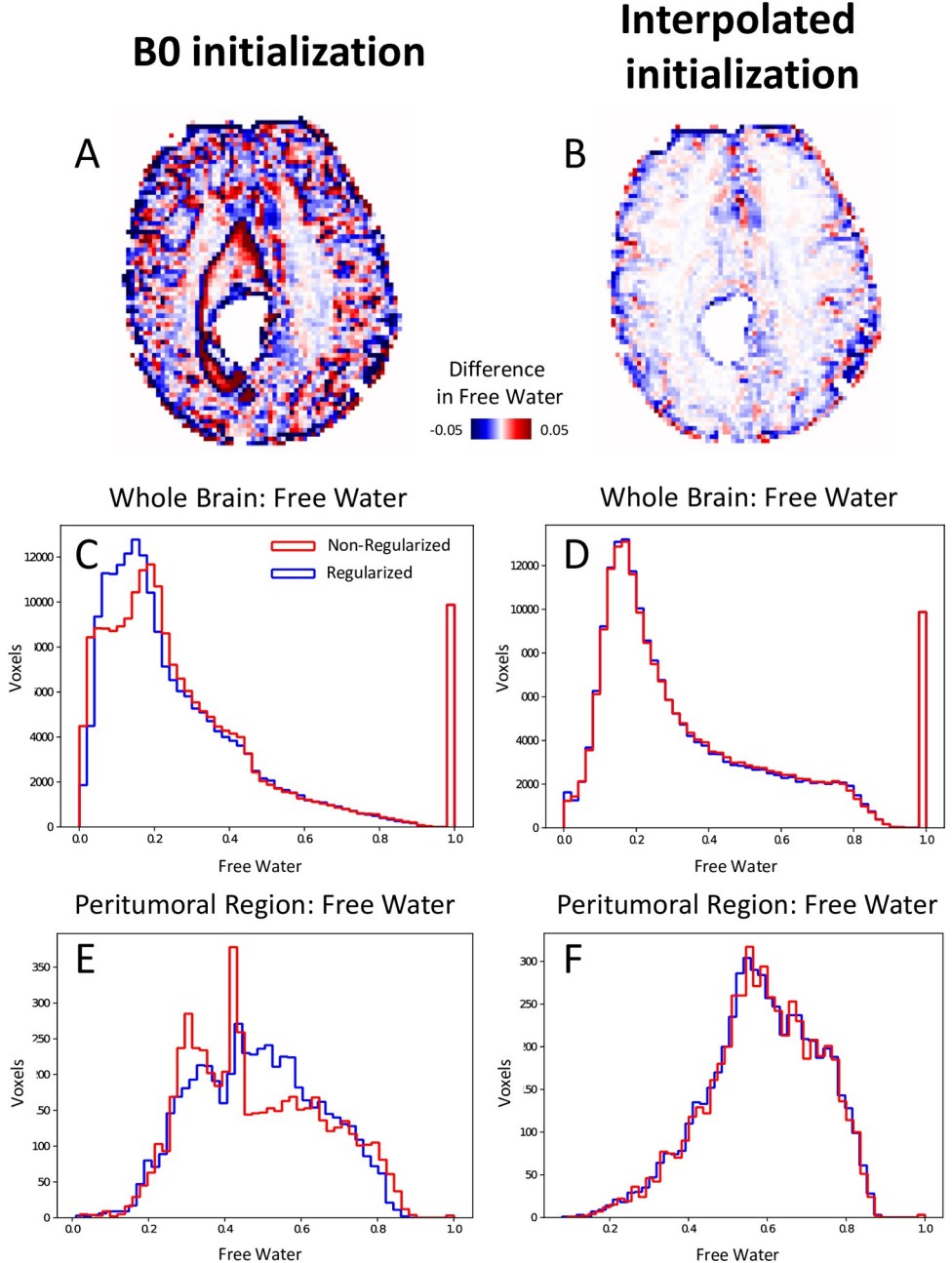

**Fig 3. Effect of regularization in the optimization phase after FERNET and b0 initialization in a tumor patient.** A and B show the difference images of free water volume fraction between regularized and non-regularized fits of the b0 initialization method and FERNET, respectively. Histograms of free water volume fraction in the whole brain with and without regularization using b0 initialization differ greatly (C), while the histograms of free water using FERNET closely match (D). Histograms in E and F demonstrate a similar effect when limited to the peritumoral region.

## 3.6. Tractography

Fig 6 shows the effect on tractography as a result of free water elimination in a large dataset of 143 brain tumor patients (*Dataset 2*). Fig 6A is an example case, showing a comparison of the arcuate fasciculus between standard tensor-based tracking and FERNET-based tracking. In extending this tractography to all the patients, we found that the tractography based on

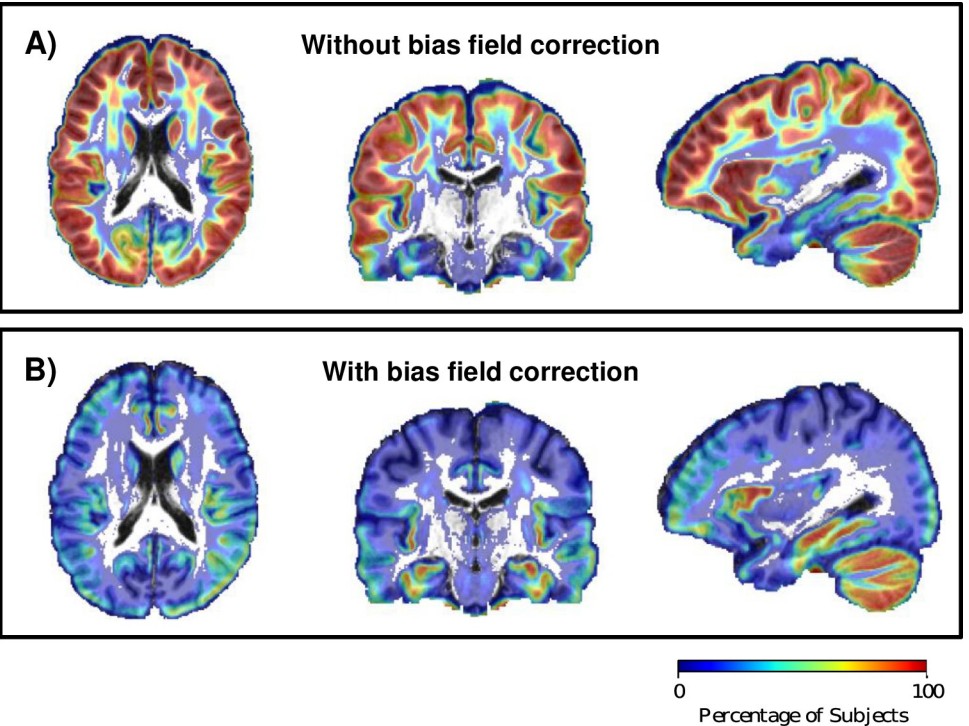

**Fig 4. Effect of bias field correction on corrected diffusion indices and free water volume fraction.** Evaluation of interpolated initialization on a large dataset (Dataset 2) of healthy controls showed that A) data without bias field correction produced more physiologically implausible voxels (defined as corrected MD $< 0.4\text{x}10^{-3}$ mm$^2$/s) than data with bias field correction (B), especially in GM regions and in the cerebellum, which are the areas impacted most by bias. The color bar represents the percentage of controls with physiologically implausible fits. Voxels with 0% of implausible fits are not plotted with color, so that the underlay template T1 image can be seen.

FERNET covered more of the peritumoral region than tractography using the standard tensor fit. This is depicted in Fig 6B by a histogram of percent difference in coverage of the edematous region by tracts. In a majority of patients, the percent difference in edema coverage was positive, indicating that tractography on free-water-corrected tensor maps traveled through a larger portion of the peritumoral region.

## 4. Discussion

We have introduced a novel initialization strategy, FERNET, for free water estimation in single-shell clinically feasible diffusion MRI, aimed at improving the characterization of peritumoral edema. The efficacy and wide applicability of our method in estimating free water volume fraction has been demonstrated comprehensively both on simulated data representing varying levels of edema and underlying tissue with and without infiltration, as well as datasets of healthy controls and patients with diverse types of brain tumors. We show that the novel strategy improves the accuracy of free water estimation for peritumoral regions. We demonstrate its applicability to different acquisitions, especially clinically feasible DTI data, underlining the widespread generalizability of our method as compared to other multicompartment modeling methods, some of which require specific acquisitions infeasible in the clinic.

Most free water elimination methods to date have been designed for healthy tissue. Developing a method for a pathology, like brain tumors, is challenging due to the absence of ground truth. We addressed this problem by generating simulated data representing edema using

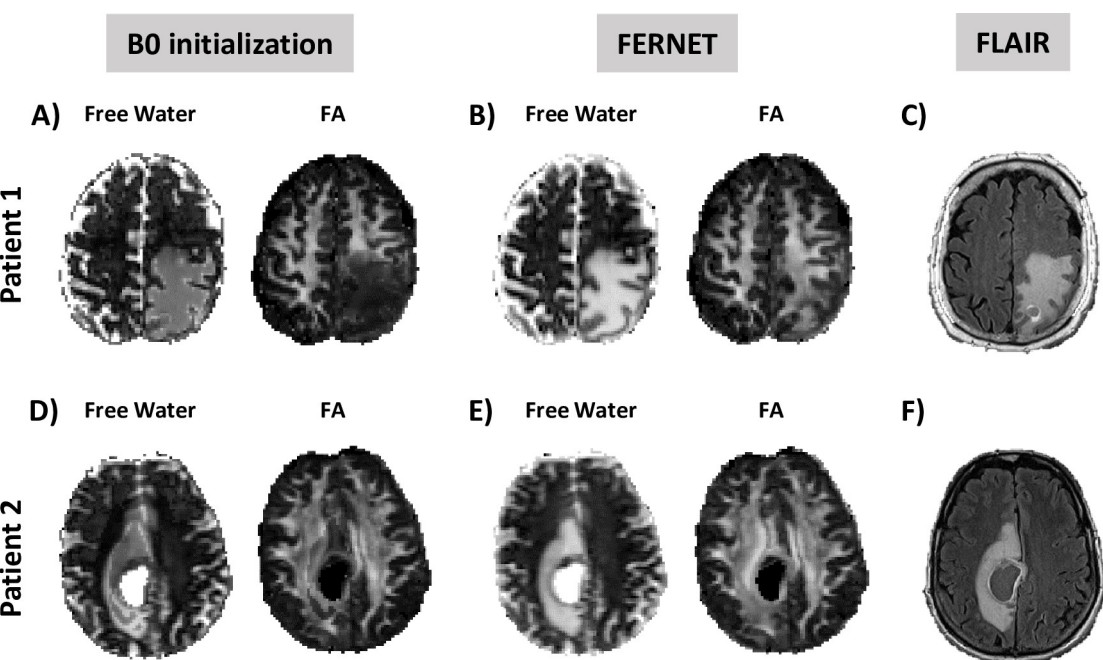

**Fig 5. Comparison of free water estimation and corrected FA maps between the two initialization methods in two brain tumor patients.** Patient 1 is a metastatic brain tumor patient and Patient 2 is a glioblastoma patient. Corrected FA maps obtained with FERNET (B, E) show better agreement between the peritumoral region and the contralateral WM compared to b0 initialization (A, D). Free water volume fraction maps obtained with FERNET are spatially smoother in the peritumoral region. This is likely due to the change in heuristics when the initial map is outside of the range of $[f_{min}, f_{max}]$. Free water volume fraction maps of FERNET more closely resemble the hyperintense region of the FLAIR maps (C, F) in the same patients at approximately the same location.

different ground truth diffusivities, anisotropies, and free water volume fractions, and tested our proposed interpolated initialization against the existing b0 initialization method. Fig 1 highlights the significant impact of the initialization strategy on the free water estimation using three simulated clinically relevant scenarios. The findings show that the proposed initialization strategy yields improved free water estimation, both in healthy tissue as well as tumor tissue with restricted diffusivity, that were affected by simulated "edema". Also, we showed that FERNET's interpolation approach provides a robust estimation that is less susceptible to noise. Although, as expected, the free water estimation became increasingly robust as SNR increased, the effect of SNR on the mean of free water estimation was negligible beyond an SNR of 20.

The success of FERNET in simulated scenarios motivated its application to human data (Fig 2). However, ground truth in human data is impossible to ascertain *in vivo*, especially when tissue is contaminated with edema. While there are several estimation paradigms, we used the free water estimated by Hoy et al. [28] as a reference standard. The estimation in this method relies on multi-shell data and claims to have a closed form solution to a problem that is ill-posed when attempted using single-shell data. The voxel-wise correlation and MSE maps show that FERNET based free water maps are more similar in signal contrast to the reference standard, and correlate better with the free water estimation of this advanced method compared to b0 initialization. This increased agreement of FERNET, suggests its superiority in the peritumoral region and also in healthy tissue. There are other multi-shell modeling methods available which can estimate fraction of various compartments, such as NODDI [33]. NODDI is a complex multi-compartment model which includes an isotropic free water compartment,

## A) FERNET-based Tractography

**Standard Tensor Fit** **FERNET** **Overlap**

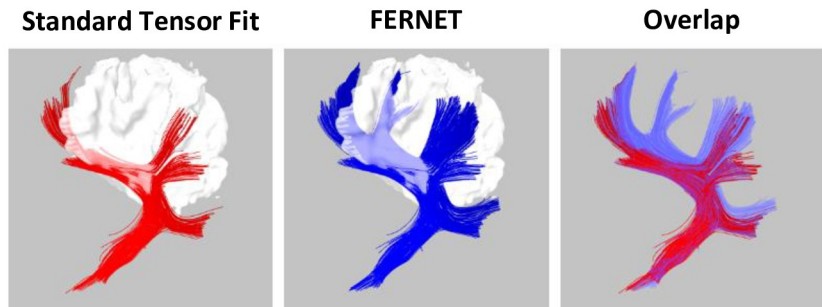

## B) Percent Difference in Edema Coverage

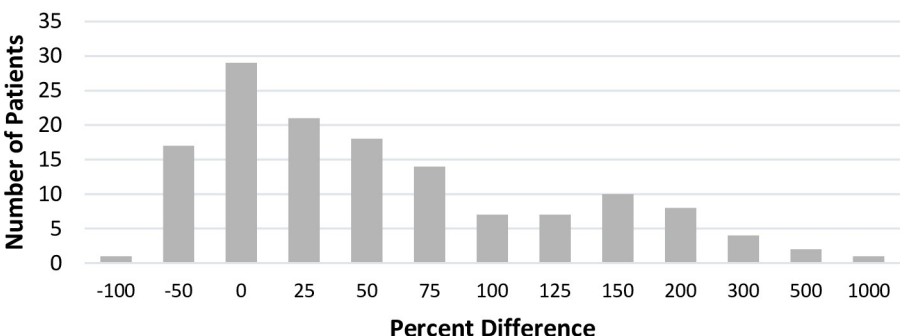

**Fig 6. Fiber tractography with and without free water elimination using FERNET and RecoBundles tract clustering.** A) Tractography of the arcuate fasciculus in one example brain tumor patient demonstrates the improvement in tracking inside the peritumoral region using free water elimination. Using standard tensor fit based tracking, streamlines appear to stop prematurely when they reach voxels affected by the accumulation of free water in the peritumoral region. The overlap of both approaches is shown, highlighting the improvement observed with FERNET. B) Tractography and reconstruction of ten major tracts using RecoBundles in 143 tumor patients showed an increase in the volume of the peritumoral region traversed by streamlines, as shown in a histogram of percent difference (relative to standard tensor-based tractography) in edema coverage. The findings demonstrate that most patients exhibit an increase in the extent of the peritumoral region covered by recognizable tracks when free water elimination is employed.

an extracellular hindered diffusion compartment, and an intracellular compartment with a fixed diffusivity. This model relies on a number of *a priori* fixed parameters based on healthy data and a specific diffusion sequence. Hence, we determined that it would not be an ideal method to compare against, opting instead for the Hoy et al [28] method, which models the same two compartments as FERNET with multi-shell data.

As free water elimination using single-shell DTI is an ill-posed problem, several implementation choices are needed with every free water elimination approach. Previous approaches (Pasternak et al. 2009) have emphasized the importance of regularization. FERNET initialization has been shown (Fig 3) to be less sensitive to regularization, implying that FERNET yields a solution that varies smoothly across neighboring voxels. On the other hand, the estimation using b0 initialization appears to be more influenced by regularization, with most impacted regions in healthy-appearing GM and the boundaries of the peritumoral region. This effect is further highlighted by the histograms of free water fraction with and without regularization. This is likely driven by the heuristic choice to replace initial estimates outside the range [$f_{min}$,

$f_{max}$] with the mean of those two values. Such a choice imposes a drastic difference between a voxel just within that range and a neighboring voxel which falls out of that range.

The importance of such implementation choices is emphasized in the FERNET paradigm, firstly, by our choice to perform bias field correction of the data. This reduced the prevalence of implausible diffusivity values in a large set of healthy controls, which were defined as having MD less than $0.4x10^{-3}$ mm$^2$/s, a value that is well below most estimates of the MD of healthy brain tissue. This finding suggests that, in the presence of magnetic field inhomogeneity in the scanner, the preprocessing of diffusion data may play a significant role in modeling the diffusion signal with this bi-compartment model, and it must be approached carefully. The second crucial implementation decision was to constrain the initial volume fraction estimate to the boundaries ($[f_{min}, f_{max}]$) and scaling the b0 image by the 5$^{th}$ and 95$^{th}$ percentile of the unweighted signals of WM and CSF. This resulted in FERNET producing a contrast in the volume fraction map that was more consistent with that obtained from FLAIR images, as compared to that produced by the b0 initialization. FLAIR is the clinical modality of choice to draw inferences about the peritumoral region, because in FLAIR images the inversion time is carefully chosen to null the signal from the CSF while increasing the signal intensity in edema and areas of inflammation with increased partial voluming of fluid. This suggests that the free water maps obtained with FERNET may have clinical relevance in locating and characterizing lesions and areas of inflammation such as in the tumor microenvironment, inflammatory plaques of multiple sclerosis, or inflammation associated with traumatic brain injury. Some subtle yet important characteristics of FERNET-based characterization are that it can be applied a dataset without needing registration to a template, or any tumor or edema segmentation.

The clinical significance of FERNET is underscored by examining the free water volume fraction maps in different types of brain tumors that may have varying degrees of edema. The findings in the peritumoral region of two brain tumor patients (Fig 5), one with a metastatic tumor and one with a diagnosis of glioblastoma, showed that the interpolated initialization allowed for reconstructing FA in the region of WM ipsilateral to the tumor, despite the presence of edema. The b0 initialization based free water estimation struggled significantly, especially in the case of metastatic brain tumor. Thus, FERNET is generalizable to peritumoral regions with different tumor types.

Finally, tractography to delineate eloquent fiber tracts has important clinical implications for the surgical treatment of malignant brain tumors [34]. Although no current fiber tractography method is completely devoid of limitations, their tract delineation capabilities make them a valuable tool for neurosurgeons [35, 36]. However, the standard tensor fit does not account for any free water, and the presence of edema leads to erroneous estimations of diffusion indices, most notably FA. This may cause some tracking algorithms to stop prematurely (Fig 6) in edematous regions, leading to spuriously distorted and/or interrupted tracts [37]. Although this is more pronounced in the presence of edema, it is important to note that even in the absence of pathology, cerebrospinal fluid partial voluming affects the reconstruction of WM tracts such as the fornix, potentially resulting in misleading inferences about brain connectivity.

The improved free water estimation that our initialization provides leads to a better modeling of brain tissue. Hence, the elimination of the confounding free water in the diffusion signal, whether it is due to partial voluming or pathology, results in better tractography than the standard single tensor model can provide. This impact on tractography was evident in the peritumoral regions of brain tumor patients with gliomas and metastases. Irrespective of the type of edema, tractography using tensors obtained from FERNET-based free water estimation was much improved, as compared to a standard tensor fit, in terms of the increase in edema coverage by the tracts after free water elimination.

There is a growing evidence that the objective of "maximal safe resection" [38, 39] is best achieved with detailed mapping of the brain tumor and surrounding white matter tracts. This emphasizes the importance of free water elimination for tractography, paving the way for robust surgical planning, and superior tracking intraoperatively [40]. FERNET provides a method that can possibly enhance clinical outcomes, due to the improvement in tractography, which is expected to have a large impact on a broad spectrum of applications that use dMRI, including neurosurgical planning [41, 42], assessing connectivity changes in pathologies traumatic brain injury [43], and stroke [44].

## Limitations

Due to the absence of the ground truth values in human data, this study relies in part on simulations derived from human diffusion data, which lack a ground truth. Simulated data representing healthy white matter was created from average values of healthy controls in WM. The ground truth free water VF of these voxels, as measured, is not known. Thus, two simulations of healthy WM were performed, representing a range of possible values. Similarly, using tumor data from one patient may not be representative of the vast heterogeneity of tumors. However, in the absence of biopsies in healthy and peritumoral tissue, these constitute exemplar scenarios over a spectrum of healthy to diseased to demonstrate FERNET's applicability. The single tensor fit of FERNET cannot distinguish between isotropic and anisotropic diffusion with high orientation dispersion, which may occur in the context of cancer, for which a multi-shell acquisition and advanced modeling would be required. Also, since it is a tensor-based model, tractography is inherently limited in the crossing fiber regions. We used deterministic tractography for evaluation, as that is used in surgical planning software, and probabilistic tractography does not perform well on tensors. However, the observed improvement in tractography in the peritumoral region, even with clinically acquired data where higher order diffusion models cannot be fitted, highlights the utility of FERNET. Finally, we since the initialization of our method performs better with bias correction of the B0 image, it relies on the bias correction tools to correct issues with diffusion data, such as susceptibility artifacts in the fronto-orbital region, or signal drop-out issues caused by image distortion. We have used the tool that is the state of the art, however, alternate approaches may also be used.

## 5. Conclusion & future work

We have designed a free water estimation paradigm based on a novel initialization approach, FERNET, that can be applied to single-shell diffusion acquisitions, readily available in the clinic. The initialization approach, which interpolates diffusivity and T2 information (b0), improves free water estimation in edematous peritumoral tissue, as well as healthy tissue, compared to traditional approaches. This is a significant contribution because it can be applied to prospective and retrospective clinical studies.

FERNET initialization for free water estimation has a myriad of potential applications in the field of diffusion MRI, especially in pathologies that provoke an increased free water, like tumors and traumatic brain injury. As it is designed to work with clinically feasible single-shell diffusion data, it can be used to improve tractography in retrospective studies using archival data, as shown by the improvement in tractography in a cohort of brain tumor patients. A comprehensive investigation of different tractography algorithms, both deterministic and probabilistic, and on more complex diffusion models in single-shell and multi-shell data, is beyond the scope of this manuscript and will be investigated in the future. The free water volume fraction map delivers valuable information about the free water content of brain tissue in the presence of pathology. This may be used in the future to help distinguish different types of

brain tumors and their genetic underpinnings based on the nature of the peritumoral tissue, paving the way for better radiomic markers of cancer.

The applicability of FERNET in clinically feasible acquisitions is expected to facilitate investigations of large tumor datasets like ABTC (http://www.abtconsortium.org) that have single shell acquisitions. The tractography tested here was based on single fiber estimation, and approaches with more complicated fiber models may assist in simultaneously resolving partial volume and complex fiber architecture in the context of peritumoral regions. Finally, an interesting application of this method can be in improved estimation of structural connectomes in the presence of edema, as the estimation of structural connections in the brain is expected to improve with a better estimation of the underlying tissue compartment after mitigating the effects of partial voluming and/or pathology. These avenues will be explored in future studies.

## Author Contributions

**Conceptualization:** Drew Parker, Abdol Aziz Ould Ismail, Simon Alexander, Wes Hodges, Ofer Pasternak, Emmanuel Caruyer, Ragini Verma.

**Data curation:** Drew Parker.

**Formal analysis:** Drew Parker, Abdol Aziz Ould Ismail, Emmanuel Caruyer.

**Funding acquisition:** Wes Hodges, Ragini Verma.

**Investigation:** Drew Parker, Ofer Pasternak, Ragini Verma.

**Methodology:** Drew Parker, Abdol Aziz Ould Ismail, Ofer Pasternak, Emmanuel Caruyer, Ragini Verma.

**Resources:** Ragini Verma.

**Software:** Drew Parker.

**Supervision:** Ragini Verma.

**Visualization:** Drew Parker.

**Writing – original draft:** Drew Parker, Abdol Aziz Ould Ismail, Ragini Verma.

**Writing – review & editing:** Drew Parker, Ronald Wolf, Steven Brem, Simon Alexander, Wes Hodges, Ofer Pasternak, Emmanuel Caruyer, Ragini Verma.

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
