## [Decision Letter · Decision Letter 0]

21 Oct 2019

PONE-D-19-22990

Freewater EstimatoR using iNtErpolated iniTialization (FERNET): Toward Accurate Estimation of Free Water in Peritumoral Region Using Single-Shell Diffusion MRI Data

PLOS ONE

Dear Dr. Verma,

Thank you for submitting your manuscript to PLOS ONE. After careful consideration, we feel that it has merit but does not fully meet PLOS ONE’s publication criteria as it currently stands. Therefore, we invite you to submit a revised version of the manuscript that addresses the points raised during the review process.

We would appreciate receiving your revised manuscript by Dec 05 2019 11:59PM. To enhance the reproducibility of your results, we recommend that if applicable you deposit your laboratory protocols in protocols.io, where a protocol can be assigned its own identifier (DOI) such that it can be cited independently in the future. For instructions see: http://journals.plos.org/plosone/s/submission-guidelines#loc-laboratory-protocols

We look forward to receiving your revised manuscript.

Kind regards,

Pew-Thian Yap

Academic Editor

PLOS ONE

Journal Requirements:

'No conflict of interest to be reported' 

We note that one or more of the authors are employed by a commercial company: Synaptive Medical Inc.

Additional Editor Comments (if provided):

Reviewers' comments:

Reviewer's Responses to Questions

**Comments to the Author**

1. Is the manuscript technically sound, and do the data support the conclusions?

Reviewer #1: Yes

Reviewer #2: Yes

2. Has the statistical analysis been performed appropriately and rigorously? 

Reviewer #1: Yes

Reviewer #2: N/A

3. Have the authors made all data underlying the findings in their manuscript fully available?

Reviewer #1: No

Reviewer #2: Yes

4. Is the manuscript presented in an intelligible fashion and written in standard English?

Reviewer #1: Yes

Reviewer #2: Yes

5. Review Comments to the Author

Reviewer #1: The work introduced a method to estimate free water volume fraction in the peritumoral region from single-shell diffusion MRI. The method aims to improve the results of a non-linear problem (two-compartment model) by using proper initialization. The manuscript is well-written and intelligible. The analyses and validations are robust and sufficient to support the authors' argument. This work could have an interesting contribution to a busy clinical setting.

There are some minor concerns that the authors should address in full:

1. The lambda_max in Eq.[4] is set to 2.5. As I understand, this value has to be smaller than d=3.0 so that the denominator is defined. Is there any reason behind the choice of 2.5, if not another value <3.0, say 2.9?

2. During preprocessing, FA and MD maps were registered to the Eve atlas using non-linear registration. Since the atlas was built from healthy adults while the data in the experiment contain tumor, what strategy was used to ensure proper alignment and prevent registration error?

3. In Fig.1, although the violin plots are informative and useful in showing the distribution of the error, it can only show one SNR value at a time (in panel A, B, and C). SNR=40 is optimistic and might not be found in some clinical data. Given the authors had results for other noise levels as well, could the heatmap of error be used with y-axis for SNR and x-axis for FW volume fraction

4. In Fig.1 A and B, the error (of FERNET) seems to increase with increased FW volume fraction, while in C, the errors seem to be stable. Is there any explanation for this observation?

5. Could the authors give some insights into why the 'bo-initialization' fails? In Fig.1 A, B, and C, besides higher error, in some cases, the distribution of the 'bo-initialization' results are scattered or skewed with outliers, which is not observed in FERNET results. Also, the trend of 'b0-initialization' results is harder to predict (overestimate in some cases but underestimate in other cases). A discussion on this would yield more insights into the problem.

6. The multi-shell ground truth used was from [18], which is a two-compartment model. I think NODDI is a better model to used as the ground truth given it also utilizes multi-shell data and has three compartments.

7. From Fig.2, seems like the FERNET always overestimates the FW volume fraction (which is consistent with Fig.1). Is there any explanation for this and potential method for correction?

8. Diffusion MRI signal suffers from the degeneracy problem that isotropic diffusion (sphere) could have a similar signal compare to anisotropic diffusion with high dispersion (e.g. random oriented sharp tensor). I expect the peritumor region would have anisotropic diffusion with high dispersion since tumors could distort fibers pathway. This would be problematic as the solver could not distinguish between true free water diffusion and high dispersion anisotropic diffusion. Is this an issue in FERNET?

9. Single-shell clinical data usually has b=800, 1000, or 1500. Is the value of b-shell affect the estimation? For instance, free diffusion signal at b=1000 or 1500 is close to zero and more likely to be affected by the Rician noise floor than signal at b=800.

Reviewer #2: The authors proposed a method to improve free water estimation (FWE) in peritumoral regions with single-shell diffusion MRI data, by better initialization. The new initialization method took advantage of the image intensity in the b0 image to find the ratio between free water and tissue water. The paper presented a relatively comprehensive validation of the FERNET method with simulation data and experimental data in both healthy brains and tumor patients. The results look convincing and the topic potentially has an important impact for tumor research. I have several comments below.

1. Methods 2.2, Equation [2], I am not this equation tells an interpolation between two initialization strategies. Is that the “x” between f_b0 and f_MD stands for interpolation? Consider change the mathematical formulation.

2. Since the FERNET heavily depends on the image intensity of the b0 image, errors can happen when b0 is problematic. 1) As the author mentioned, field inhomogeneity can affect the results and they used N4 correction. But it is know that N4 is not always successful. 2) Signal dropout due to susceptibility artifacts, e.g., in the frontal-orbital area. Please address these issues.

3. f_b0 replies on the ratios between S0, Sw (CSF), and St(white matter). How about gray matter, how was the fitting done in the GM voxels?

4. Some of the methodological details are quite empirical. For example, “we define as the 5th percentile of the unweighted signal in a region of WM, and we define as the 95th percentile of unweighted signal in a region of CSF”. How the percentiles were determined? Also, “the value of in Equation 2 is set following Equation 3”, I do not see rationale of determining this way. Equation 3 in itself is also empirical.

5. Was regularization performed in this study and how? The authors mentioned in Results 3.4, that the proposed method was less dependent on regularization. This needs to be clarified.

6. Method 2.3 “0_ and 0_ are values taken as reference from selected voxels in human data”. Where were the selected voxels? I think it is much better to do a segmentation of the b0 image to obtain S0 in WM, GM, and CSF.

7. Method 2.4.1 “Automatic tumor and peritumoral region segmentation was obtained in the patients from Dataset 1 and Dataset 2 using GLISTR”. Please access the segmentation accuracy, at least in part of the data.

8. Results 3.2, in the simulation, both b0 initialization and interpolated initialization results showed that the estimation errors increased as the ground truth free water fraction increased. Please explain.

9. Figure 2, it is counterintuitive that high correlation was shown in green, low correlation was shown in red, and negative correlation is shown in blue.

10. In Figure 5, should (A) and (B) be switched? It looks like that the percentage of unrealistic fit is higher with bias field correction. Also, some WM regions in this figure were white, which is out of the colorbar. What does that indicate?

11. To test the accuracy of fiber tracking, it would be better to do the validation with the multi-shell data and compare the results with multi-shell FWE.

6. PLOS authors have the option to publish the peer review history of their article (what does this mean?). If published, this will include your full peer review and any attached files.

Reviewer #1: No

Reviewer #2: Yes: Dan Wu

---

## [Author Response · Author response to Decision Letter 0]

29 Apr 2020

The "clean" file is in the PlosOne format 

The "tracked" file has all the changed tracked 

The figures have been checked on the software approved by PlosOne and each of them are uploaded separately 

Data has been uploaded and statement added to cover letter and the end of the paper 

Funding statement: Synaptive Medical's role in the paper was clarified.

---

## [Decision Letter · Decision Letter 1]

11 May 2020

Freewater EstimatoR using iNtErpolated iniTialization (FERNET): Characterizing Peritumoral Edema Using Clinically Feasible Diffusion MRI Data

PONE-D-19-22990R1

Dear Dr. Verma,

We are pleased to inform you that your manuscript has been judged scientifically suitable for publication and will be formally accepted for publication once it complies with all outstanding technical requirements.

With kind regards,

Pew-Thian Yap

Academic Editor

PLOS ONE

Additional Editor Comments (optional):

Reviewers' comments:

Reviewer's Responses to Questions

**Comments to the Author**

1. If the authors have adequately addressed your comments raised in a previous round of review and you feel that this manuscript is now acceptable for publication, you may indicate that here to bypass the “Comments to the Author” section, enter your conflict of interest statement in the “Confidential to Editor” section, and submit your "Accept" recommendation.

Reviewer #1: All comments have been addressed

Reviewer #2: All comments have been addressed

2. Is the manuscript technically sound, and do the data support the conclusions?

Reviewer #1: Yes

Reviewer #2: Yes

3. Has the statistical analysis been performed appropriately and rigorously? 

Reviewer #1: Yes

Reviewer #2: Yes

4. Have the authors made all data underlying the findings in their manuscript fully available?

Reviewer #1: No

Reviewer #2: No

5. Is the manuscript presented in an intelligible fashion and written in standard English?

Reviewer #1: Yes

Reviewer #2: Yes

6. Review Comments to the Author

Reviewer #1: (No Response)

Reviewer #2: The authors have adequently addressed all my comments. The mansucript is ready for publication on PLOS

However, the response to comment#3 that "DTI is primarily a WM modality ..." is not entirely true. DTI is useful in characterizing GM too, e.g., hippocampus and cortex. There's a number of studes. I would suggest the authors to explictly acknowledge that their current approach is not suitable for GM.

7. PLOS authors have the option to publish the peer review history of their article (what does this mean?). If published, this will include your full peer review and any attached files.

Reviewer #1: No

Reviewer #2: Yes: Dan Wu

---

## [Editor Report · Acceptance letter]

15 May 2020

PONE-D-19-22990R1 

Freewater EstimatoR using iNtErpolated iniTialization (FERNET): Characterizing Peritumoral Edema Using Clinically Feasible Diffusion MRI Data 

Dear Dr. Verma:

I am pleased to inform you that your manuscript has been deemed suitable for publication in PLOS ONE. Congratulations! Your manuscript is now with our production department. 

With kind regards,

on behalf of

Dr. Pew-Thian Yap 

Academic Editor

PLOS ONE